# International assessment of the link between COVID-19 related attitudes, concerns and behaviours in relation to public health policies: optimising policy strategies to improve health, economic and quality of life outcomes (the iCARE Study)

Simon L Bacon [1,2] Kim L Lavoie,[1,3] Jacqueline Boyle,[4,5] Jovana Stojanovic,[1,2] Keven Joyal-Desmarais [1,2] for the iCARE study team[1]

► Prepublication history and additional materials for this paper is available online. To view these files, please visit the journal online (http://dx.doi.org/10.1136/bmjopen-2020-046127).

For numbered affiliations see end of article.

**Correspondence to**
Dr Simon L Bacon;
simon.bacon@concordia.ca

## ABSTRACT

**Introduction** In the context of a highly contagious virus with only recently approved vaccines and no cure, the key to slowing the spread of the COVID-19 disease and successfully transitioning through the phases of the pandemic, including vaccine uptake, is public adherence to rapidly evolving behaviour-based public health policies. The overall objective of the iCARE Study is to assess public awareness, attitudes, concerns and behavioural responses to COVID-19 public health policies, and their impacts, on people around the world and to link behavioural survey data with policy, mobility and case data to provide behavioural science, data-driven recommendations to governments on how to optimise current policy strategies to reduce the impact of the COVID-19 pandemic.

**Methods and analyses** The iCARE study (www.icarestudy.com) uses a multiple cross-sectional survey design to capture self-reported information on a variety of COVID-19 related variables from individuals around the globe. Survey data are captured using two data capture methods: convenience and representative sampling. These data are then linked to open access data for policies, cases and population movement.

**Ethics and dissemination** The primary ethical approval was obtained from the coordinating site, the Centre intégré universitaire de santé et de services sociaux du Nord-de-l'Île-de-Montréal (REB#: 2020–2099/03–25–2020). This study will provide high-quality, accelerated and real-time evidence to help us understand the effectiveness of evolving country-level policies and communication strategies to reduce the spread of the COVID-19. Due to the urgency of the pandemic, results will be disseminated in a variety of ways, including policy briefs, social media posts, press releases and through regular scientific methods.

## Strengths and limitations of this study

► This is a large, international study that has data captured from over 150 countries.
► The survey was constructed around well-recognised behavioural theories and frameworks.
► The study is primarily being conducted online that may limit some of the generalisability of the data that are available, especially in lower and middle income countries.
► The primary data capture method is through snowball sampling, which is likely to create some bias in the sample. However, some of this can be adjusted using weightings from the representative samples that are being collected.
► A key strength of the study is that it has been developed to provide constructive policy and communication strategy data that can be implemented by governments to improve adherence to COVID-19 mitigation methods.

## INTRODUCTION

With only recently approved vaccines and no cure, the key to slowing the spread of COVID-19 and successfully transitioning through the phases of the pandemic, is *public adherence* to unprecedented and rapidly evolving behaviour-based public health policies.[1 2] To date, adherence to these policies has been critical to reducing the spread of COVID-19 and have ranged from personal hygiene measures (eg, hand washing) to strict lockdown measures (eg, business and school closures).[3–5] However, adherence to most of these policies requires making behavioural changes that may come with significant personal, social and economic costs, which may undermine their impact.[6] For example, despite

public health messages promoting the 'advantages' of adhering to COVID-19 mitigation measures, adherence to policies that may come with high personal costs (ie, physical distancing) have been much poorer (54%) than for other 'less costly' behaviours like hand washing (90%).[7] Furthermore, as we look towards changing lockdown measures, people's willingness to adhere to evolving government recommendations (eg, school and store reopenings and receiving vaccines) will also be critical for re-engaging the economy while minimising the potential for future waves of the pandemic. Unfortunately, policy variations between and within countries have created public confusion and uncertainty about government policy motives.[8] In addition, governments have predominantly designed policies based on how they believe people 'should' behave and have ascribed little consideration to what we know about how people actually behave.[9 10]

Decades of behavioural science research has revealed that human behaviour is predictable and modifiable.[11] Multiple factors are likely to predict why people adhere

(or not) to various public health measures which, in the context of COVID-19, can be defined using two related behaviour prediction models: (1) *The Capability, Opportunity, Motivation-Behaviour (COM-B) Model*,[2 12] which predicts that behaviour change depends on: awareness of prevention measures and the ability to enact them (capability), the belief that measures are personally relevant and important (motivation) and having the social and environmental resources required to adopt the behaviour (opportunity) (see figure 1A) and (2) *The Health Beliefs Model*,[13 14] which posits that in adopting disease prevention measures, a person's belief in the personal threat(s) posed by the disease, together with a person's belief in the importance and effectiveness of recommended behaviours, will predict the likelihood a person adopting (or not) a particular behaviour (figure 1B). In the context of this unprecedented health, social and economic crisis, where the global need for adherence to rapidly evolving public health policies has never been greater, our understanding of the determinants of adherence at each

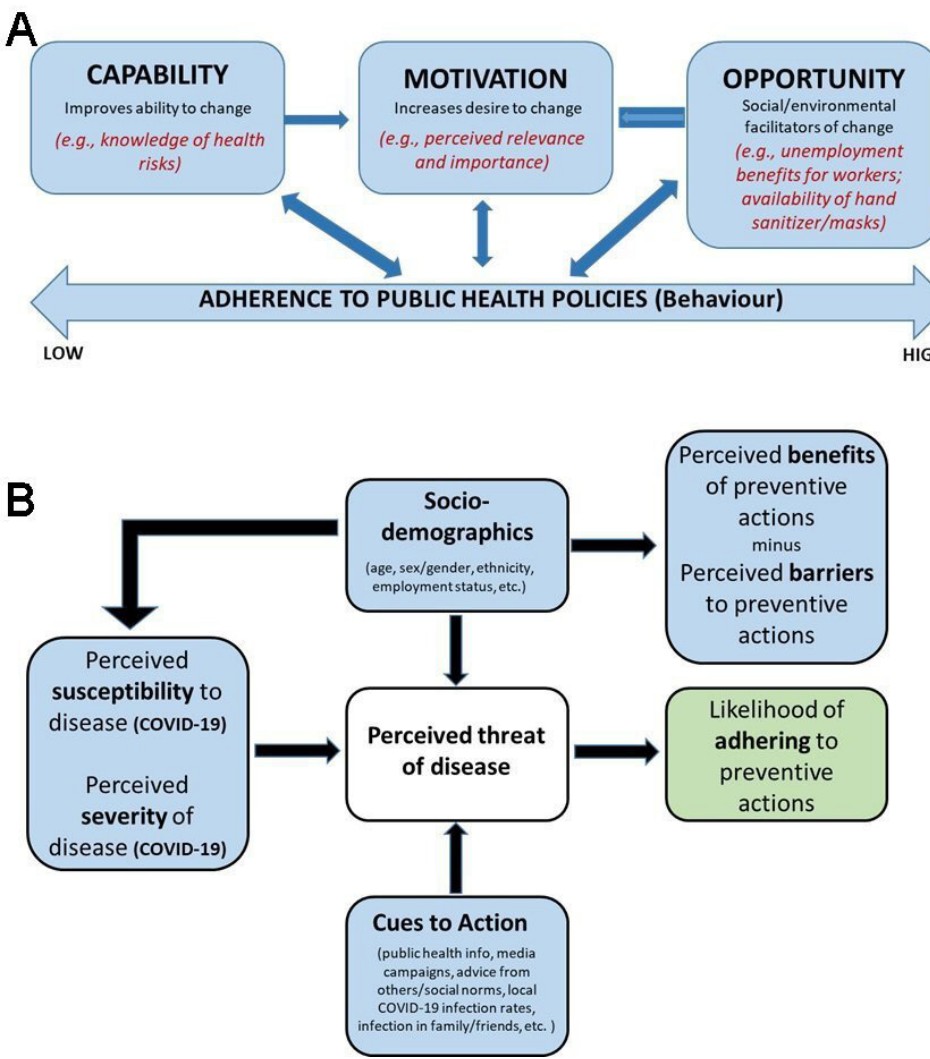

**Figure 1** The theoretical models underpinning the behavioural responses to COVID-19.

phase of the pandemic, and as a function of various policies, is critical for effective policy planning, communication and effectiveness.

The overall goal of the iCARE Study is to assess public awareness, attitudes, concerns and behavioural responses to COVID-19 public health policies, and their impacts, on people around the world (www.icarestudy.com), and to link behavioural survey data with policy, mobility and case data to provide behavioural science, data-driven recommendations to governments on how to optimise current policy strategies to reduce the impact of the COVID-19 pandemic worldwide. Specifically, we will address the following:

1. What are the key individual characteristics (eg, socio-demographic, psychological, behavioural, physical/mental health and economic) that are associated with adherence to major COVID-19 public health policies *in general* and by *country*?
2. To what extent are COVID-19 attitudes, beliefs and concerns associated with adherence, and how does this vary across key subgroups (eg, age, sex, income, family/household structure, ethnic groups and those with health conditions)?
3. What are the short-term and medium-term impacts of COVID-19 and its public health policies, and how do they vary as a function of key individual characteristics in *general* and by *country*?
4. Which policies and strategies are associated with better (and worse) adherence, are most (and least) effective at reducing infection rates and positively impact economic growth (where appropriate)? As well as identifying in whom these polices and strategies worked (and did not work).
5. The development of behavioural science, data-driven, tailored recommendations that governments could use to optimise policy and communication strategies to improve adherence, as well as health, economic and quality of life outcomes.

## METHODS AND ANALYSIS
### Study design
The iCARE Study is a Canadian-led, ongoing, multiwave international study involving the collaboration of more than 190 international researchers from over 40 countries (see online supplemental material). It uses a multiple cross-sectional survey design (each approximately 6 weeks apart) to capture self-reported information on a variety of COVID-19 related variables from individuals around the globe. Survey data are captured using two data capture methods: convenience and representative sampling (see details further). These data are then coupled to open access data for policies, cases and population movement. The study is managed by the Montreal Behavioural Medicine Centre (MBMC; a joint Centre intégré universitaire de santé et de services sociaux du Nord-de-l'Île-de-Montréal (CIUSSS-NIM)/Université du Québec à Montréal/Concordia University academic research and training centre).

### Patient and public involvement (PPI) statement
Given the significance and broad impact of the COVID-19 pandemic, PPI is crucial for effective research in this area. More importantly, given the global nature of the iCARE study, it has been critical to have individuals from multiple settings included in the development of the various elements and items in the survey. To this end, we consulted with over 190 multi-disciplinary collaborators (including experts from the behavioural sciences, medicine and infectious disease, public health, epidemiology, statistics and implementation science) from more than 40 countries including researchers, clinicians, students and members of the general public in the development and design of the iCARE study (see online supplemental material for the iCARE team). In addition, throughout our data analysis process, we have engaged critical end users, including government officials, the public and the news media, in defining areas that need critical input for which the iCARE study is able to address.

### The iCARE survey
The core elements of the survey assess the following domains:
▶ Awareness of local COVID-19 public health policies.
▶ Attitudes/beliefs about local COVID-19 policies.
▶ Behavioural responses to local COVID-19 policies.
▶ Perceived concerns about COVID-19.
▶ The impacts of COVID-19 and its policies (social, occupational, economic, quality of life, physical and mental health).
▶ COVID-19 information sources.
▶ COVID-19 testing and infection status.
▶ Impacts on schools and education.
▶ Physical and mental health status.
▶ General health behaviours, including vaccine history, attitudes and behaviours.
▶ Sociodemographics and socioeconomic barriers and facilitators of adherence.

Most questions are aligned with the constructs in both the COM-B (see figure 2)[12] and Health Belief Models.[13][14] Questions assessing COVID-19 impacts were also chosen to facilitate data harmonisation with international COVID-19 studies involving the National Institutes of Health (NIH) and WHO.[15] The survey is currently available in 36 languages, making it legible to the majority of the world's population.

Though the core content of the survey is consistent throughout each release cycle, small modifications have been made as a function of the evolving nature of COVID-19 and public health policies. All surveys are open access and can be found at: https://osf.io/nswcm. Regardless of the survey content, each questionnaire is designed to take no more than 15–20 min to complete.

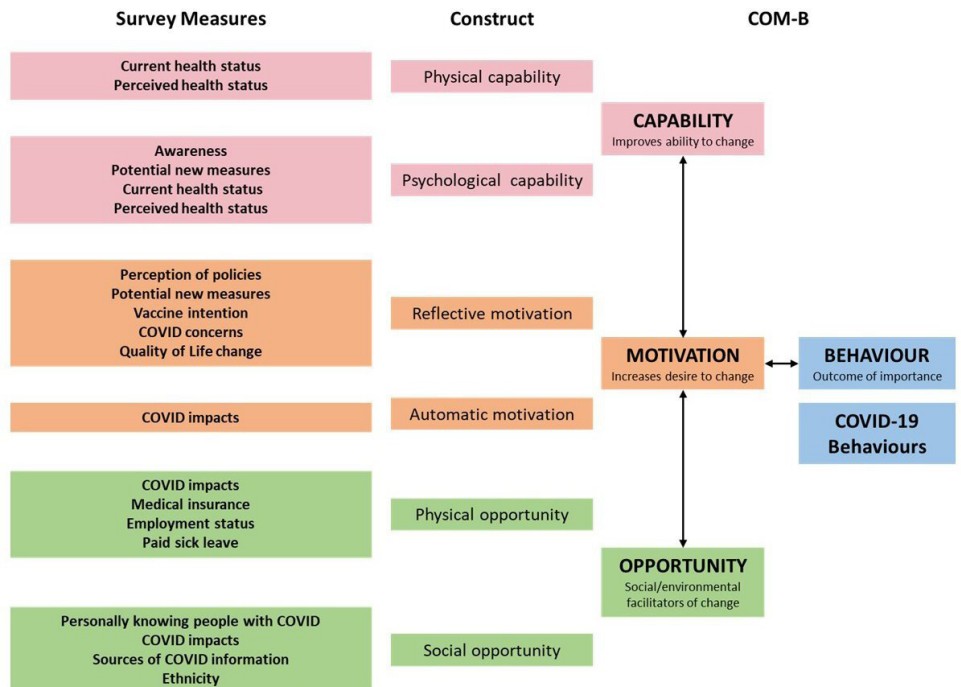

**Figure 2** Measures within survey 2 mapped onto the COM-B model. (1) This model is conceptual and needs to be tested. (2) For a number of items, we are using COVID-19 behaviour implicitly rather than explicitly. For example, impact of COVID-19 might not be due directly to the COVID-19 behaviour but are expected to be indirectly related to COVID-19 behaviours, which is not consistent with a 'pure' COM-B definition, and (3) several items may overlap with more than one components of COM-B depending on interpretations. COM-B, Capability, Opportunity, Motivation-Behaviour.

### Global convenience sample

Survey participants are being recruited using online snowball sampling by all global collaborators. The online survey (LimeSurvey) is distributed through various channels to reach as many people around the world as possible. These channels include professional networks, associations and societies; community organisations; schools and universities; hospitals and health networks; via social media; and personal contacts. The central study coordination group creates a variety of email, social media and public facing materials for each survey round that are then translated and provided to each collaborator. There are also a series of instructional tools that collaborators can use that provide information and examples of ways in which they can distribute the survey through their local country networks.

To date, there have been seven survey releases (April, May, June, July, September, November and December).

There are several current funding applications that are being reviewed, which if funded, would extend the data collection to eight more releases through to January 2022 (see figure 3).

### Representative samples in target countries

To supplement convenience sampling, we have been conducting parallel national representative sampling in countries where funds are available. Participants in each representative sample are balanced according to age, sex, province/region, education level and income to ensure representation across these relevant variables. Representative sampling uses polling services to distribute the iCARE survey, generally with internet-based sampling methods, though for certain countries, especially low-income and middle-income countries (LMICs), there may be a need to conduct telephone and in-person interviews. For example, in Canada, we have used Leger polling services,

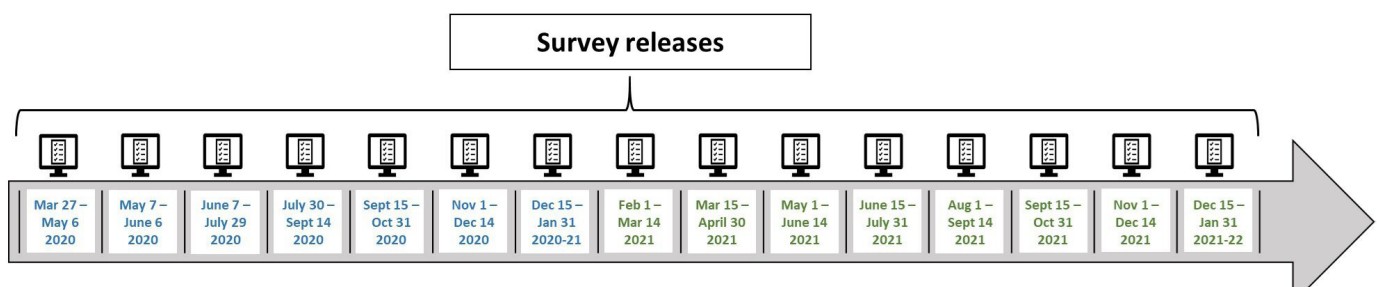

**Figure 3** Survey release timeline. Dates in blue already have funding. Dates in green are pending current funding applications.

who recruit participants aged 18 years and over through their Léo online panel (LégerWeb.com). This panel includes over 400 000 Canadians, most of whom (60%) have been recruited within the past 10 years. Two-thirds of the panel were recruited randomly by telephone, with the remainder recruited via publicity and social media. Using data from Statistics Canada, results are weighted within each province according to the sex and age of the respondents in order to make their profiles representative of the actual population within each Canadian province. Then, the weight of each province is adjusted to make it representative of their actual weight within the Canadian federation. Representative sampling in targeted countries will enable global coverage of most geographical locations and socioeconomic gradients. In addition, representative sampling will also allow us to estimate potential biases in the convenience sample data for those countries.

### Additional data sources

The Oxford COVID-19 Government Response Tracker[16 17] systematically collects publicly available information on a variety of indicators of COVID-19 related government policy responses. These policies are then accumulated to provide a variety of indexes as estimates of the total response of an individual country. Google Mobility Data[18] provides user mobility trends over time by country and region across different categories of places (eg, retail, groceries, parks, transit stations, workplaces and residential) and generates regular 'Community Mobility Reports' presented by location. They report the per cent change in visits to places like grocery stores and parks within a geographic area. These datasets show how visits and length of stay at different places change compared with baseline. Datasets show trends over several months with the most recent data reflecting the last 2–3 days. Johns Hopkins Coronavirus Resource Center[19 20] has been tracking country-level (and province/state for Canada and the USA) case, death and recovery data since the start of the pandemic, and the website is updated multiple times a day. In addition, they provide testing data for US states. The data are drawn from multiple sites.

### Data harmonisation

Initially, all data sources will be aggregated at the country level, as a function of available data. However, for those with limited data, it might be at the level of continent, and for those with large amounts of data, we may also be able to provide data at the level of region. Data sources will be tagged based on the date when each participant completed the survey. A series of generalised linear models will be developed to estimate systematic differences in responses between sexes, ethnicities, age groups, essential worker status and other key sociodemographic variables. Patterns of missing data will be examined and, where appropriate, accounted for by using multiple imputation techniques.[21 22] In countries where there is sufficient data in the convenience sample, we will apply weights to allow the data to provide national approximations.[23–25]

### Statistical analyses

With a study of this magnitude, it is impossible to detail all possible analyses that could be conducted, as these will vary based on the specific questions that might be received from governments or researcher partners. However, the following section provides a high-level overview of the kinds of 'basic' analytical strategies that will be conducted with the data. Descriptive analyses, including general linear models or logistic regressions, of the survey data will be provided to explore trends in the main areas represented in the survey. Where possible, the psychometric properties of the various elements of the survey will be explored. This will also include a variety of clustering techniques, for example, principal components analyses (PCA) or factor analyses, to create appropriate subscales. For instance, to cluster and reduce the dimensionality of the COVID-19 impact questions for surveys 2–4, we performed a PCA on the polychoric correlation matrix of the COVID-19 impacts variables. We used an orthogonal (varimax) rotation in order to distribute the component loadings. We identified different impact components based on the Kaiser criterion (eigenvalue >1.0),[26] scree plot, component loadings (>0.4) and components interpretability. For the main study questions (see above), with the magnitude and complexity of the data that is being captured, a number of different multilevel modelling techniques will be used. As an example, exploratory iterative generalised least squares[27] models followed by Markov chain Monte Carlo estimation for some models will likely be used.[28] Briefly, this is a Bayesian simulation approach that (after assigning starting values and prior distributions) sequentially samples subsets of parameters from their conditional posterior distributions given current values of the other parameters. This is a very flexible approach used by other groups with comparable data (eg, NCD-RisC[29]). For instance, using this approach, we are evaluating how the perception of government recommendations and the population's behaviour regarding face masks wearing varies according to the date of policy implementation in five targeted countries (Canada, USA, Colombia, Brazil and France) and how this then tracks onto case rates.

For the representative samples, appropriate link functions will be tested and used, with the polling company's sampling weights being employed.[23–25] All the national representative data will leverage the global data by pooling all the available information (at any given point in time) and extending our models into a multilevel framework with random effects (intercepts and slopes) at the country levels. By essentially borrowing information from the other countries, this approach will improve the power to obtain robust and precise estimates for any singular country.[23 30] In addition, where possible, we will leverage the representative samples to be able to validate the 'representativeness' of the data captured in the global sample. These analyses may provide insights into potential areas of bias and so that potential further weightings could be applied to the global sample.

## Progress to date

### Convenience sampling

Survey 1 of the global convenience sample began on 27 March 2020. When it closed on 6 May, we had received surveys from 28 651 people in 137 countries, including more than 1000 responses from four countries and more than 500 responses from 10 additional countries. Survey 2 of the global convenience sample was launched on 5 May 2020. When it closed on 8 June, we had received surveys from 12 576 people in 124 countries, including more than 500 responses from seven countries. Survey 3 of the global convenience sample was launched on 8 June 2020. When it closed on 22 July, we had received surveys from 7652 people in 100 countries, including more than 500 responses from three countries. Survey 4 of the global convenience sample was launched on 22 July 2020. When it closed on 15 September 2020, we had received surveys from 4102 people in 81 countries, including more than 500 responses from two countries. Survey 5 of the global convenience sample was launched on 15 September 2020. When it closed on 3 November 2020, we had received surveys from 3404 people in 87 countries, including more than 500 responses from two countries. Survey 6 of the global convenience sample was launched on 3 November 2020. When it closed on 15 December 2020, we had received surveys from 2451 people in 73 countries, including more than 500 responses from one country.

### Representative sampling

To date, seven rounds of representative sampling have been captured. Three of these have occurred in Canada (survey 1: 9–20 April, n=3003; survey 3: 4–17 June, n=3005; and survey 6: 28 October–10 November, n=3005), two in Australia (survey 2: 1–5 May, n=1005 and survey 3: 1–7 July, n=1051) and one each in the UK (survey 1: 3–30 April, n=2056) and Ireland (survey 3: 22 June–15 July 2020, n=1000). Current funding will allow us to capture another two samples in Canada along with samples from the USA, Italy and Colombia. Additional samples will be captured dependent on funding.

## Ethics and dissemination

### Ethics approval

The Research Ethics Board (REB) at the coordinating study site CIUSSS-NIM provides the primary ethical approval (REB#: 2020–2099/03–25–2020). Online consent is provided by participants prior to completing the survey. No personal identifying information is collected from any participant. In addition, several of the collaborating sites have also obtained ethical approval to distribute the survey within their country or institution, though this is not required.

### Knowledge translation

Due to the evolving nature of the COVID-19 pandemic, outputs from analyses will be disseminated in a variety of ways. Regular updates will be posted to the iCARE website (www.icarestudy.com) and disseminated through the MBMC social media outlets (https://www.facebook.com/CMCMMBMC; https://twitter.com/mbmc_cmcm; https://www.instagram.com/mbmc_cmcm/). Where appropriate, press releases and news media will be targeted. Of note, our study has already received a great deal of media attention, with more than 75 print, radio and television interviews across the globe (as of 20 October 2020; see https://mbmc-cmcm.ca/COVID-19/media/ for full coverage). Within Canada, we are partnering with the Royal Society of Canada's COVID-19 Task Force to reach the general public, government and national media. Finally, we will also release results through traditional scientific methods, for example, journal articles and conference presentations. For example, survey 1 data were presented at the International Behavioural Trials Network Global 2020 Virtual meeting (see https://www.ibtnetwork.org/conference/virtual2020/video-session-2/).

### Interpretation

This study will provide high-quality, accelerated and real-time evidence to help us understand the differing impacts of COVID-19 policies, strategies and communication around the world. It will provide evidence for the effectiveness of evolving policies implemented to reduce the spread of the virus, both in general and among key subgroups (eg, younger vs older, ethnic minorities and those with health conditions). The study will also generate evolving evidence to support public health planning, decision making and responses around the world, including LMICs. Examples of the results to date can be found at https://mbmc-cmcm.ca/COVID-19/research/stats/ and https://mbmc-cmcm.ca/COVID-19/research/infog/. Of note, the iCARE study has provided data to the Canadian (Federal), Irish, Province of Ontario (Canada) and State of Victoria (Australia) governments, covering polices ranging from facemasks, contact tracing applications and COVID-19 vaccine uptake.

### Limitations

The main limitation of the study is that the survey is being conducted online. Though there is generally good internet access for most high-income countries and even some LMICs (eg, India), some LMICs have limited access in certain areas and within certain population subgroups. This, coupled with the convenience sampling method, means that there may be some degree of sample bias. Though some of this can be adjusted for based on the representative sampling data, it cannot be eliminated completely. Moreover, the fact that the iCARE survey is available in 36 languages means that certain marginalised groups (eg, immigrants to certain countries, like Canada, the USA and France, which are highly represented) will likely be able to complete the survey in their native language. This may help increase participation among those who might otherwise be excluded due to language barriers. Another limitation is the fact that we

will be conducting correlation analyses. Though we will be using some sophisticated analytical modelling, we cannot derive direct causative relationships from the study. However, our main interest is in temporal changes in attitudes and behaviours as the pandemic evolves, so analysing repeated cross-sectional cohorts still allows us to meet our study objectives.

## Conclusion

Ultimately, this study will help us understand what public health policies and strategies are working, where and for whom, which can inform changes (improvements) in policy strategy and communication to help mitigate the spread of COVID-19, especially as countries are now starting to cycle through various waves of the pandemic, and its physical/mental health, social, economic and quality of life impacts.

**Author affiliations**
¹Montreal Behavioural Medicine Centre, CIUSSS du Nord-de-l'Ile-de-Montreal, Montreal, Québec, Canada
²Department of Health, Kinesiology, and Applied Physiology, Concordia University, Montreal, Québec, Canada
³Department of Psychology, Université du Québec à Montréal, Montreal, Québec, Canada
⁴Monash Centre for Health Research and Implementation - MCHRI, Monash University, Clayton, Victoria, Australia
⁵School of Public Health and Preventive Medicine, Monash University, Clayton, Victoria, Australia

**Contributors** All authors contributed to the manuscript including: contributing substantially to conception and design of the study; drafting the article and revising it critically for important intellectual content; providing final approval of the version to be published; and acting as guarantors of the work.

**Funding** The primary source of funding for the iCARE study has been primarily through redirected funding associated with Montreal Behavioural Medicine Centre, including funds from a Canadian Institutes of Health Research-Strategy for Patient Oriented Research Mentoring Chair (SMC-151518, PI: SB), a Fonds de Recherche du Québec: Santé Chair (251618, PI: SB), a Université du Québec à Montréal Research Chair (1471, PI: KLL) and Fonds de Recherche du Québec: Santé Senior Research Award (34757, PI: KLL). The Canadian representative sampling will be funded by the Canadian Institutes of Health Research (MS3-173099, PI: SB) and the Fonds de Recherche du Québec: Société et Culture (2019-SE1-252541, PI: SB). The Australian representative sampling was funded by Monash University and indirectly by the National Health and Medical Research Council and the Medical Research Future Fund (2579, PIs: Drs Helena Teede and JB). The Irish representative sampling was funded by the Health Research Board and the Irish Research Council (COV19-2020-097, PI: Dr Gerard J Molloy). The UK representative sampling was funded by CALIBRE research funding, provided by Loughborough University, UK (5705, PI: Dr Nicola J Paine). None of the funders were involved in the study design.

**Competing interests** SB has received consultancy fees from Merck for the development of behavior change continuing education modules, speaker fees from Novartis and Janssen and has served on advisory boards for Bayer, Sanofi and Sojecci Inc, none of which are related to the current article. KLL has served on the advisory board for Schering-Plough, Takeda, AbbVie, Almirall, Janssen, GSK, Boehringer Ingelheim (BI) and Sojecci Inc and has received sponsorship for investigator-generated research grants from GlaxoSmithKline (GSK) and AbbVie, speaker fees from GSK, Astra-Zeneca, Astellas, Novartis, Takeda, AbbVie, Merck, Boehringer Ingelheim, Bayer, Pfizer and Air Liquide, and support for educational materials from Merck, none of which are related to the current article. SB, JS and KJ-D have no competing interests to declare.

**Patient consent for publication** Not required.

**Provenance and peer review** Not commissioned; externally peer reviewed.

**ORCID iDs**
Simon L Bacon http://orcid.org/0000-0001-7075-0358
Keven Joyal-Desmarais http://orcid.org/0000-0003-0657-8367

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
