## [Reviewer comments · BMJ Open]

ARTICLE DETAILS

TITLE (PROVISIONAL)	An international assessment of the link between COVID-19-related attitudes, concerns and behaviours in relation to public health policies: Optimising policy strategies to improve health, economic and quality of life outcomes (the iCARE Study). Protocol Paper
AUTHORS	Bacon, Simon; Lavoie, Kim; Boyle, Jacqueline; Stojanovic, Jovana; Joyal-Desmarais, Keven

VERSION 1 – REVIEW

REVIEWER	Viet-Phuong La Phenikaa University, Hanoi, Vietnam
REVIEW RETURNED	09-Dec-2020

GENERAL COMMENTS	The authors of this paper present a study protocols for an international assessment of the link between COVID-19-related attitudes, concerns and behaviours in relation to public health policies. I have a few suggestions that you might want to consider: - The paper should provide technical validation methods which will be used in study- The Figure 1b seems lack of the block connect to “Likelihood of adhering”?- The survey documents at https://doi.org/10.17605/OSF.IO/H8RW2 was deleted. Thank you for the opportunity to review the paper.
---

REVIEWER	MOATTI JEAN PAUL Emeritus Professor of Health Economics IMERA Aix Marseille University
REVIEW RETURNED	09-Dec-2020

GENERAL COMMENTS	In the case of HIV infection, thousands of Knowledge-Attitudes-Beliefs & Practices (KABP) surveys have been carried out worldwide in representative samples of the general population of multiple countries at various points in time and have been often very useful to guide public health authorities tailoring appropriate prevention messages and policies. The ICARE study is a Canadian-initiated international cross-sectional KABP survey in the context of the on-going pandemic of COVID-19 with the aim of allowing comparison at multiple points in time (through sequential waves) between samples in circa 140 countries. The present paper deals with the study protocol as well as some very preliminary data about the ongoing data collection process. However, there are a number of limitations in the study protocol that raises doubts about the capacity of future analyses to fulfill this complex goal of valid multi-countries’ comparison with the
---

	additional ambition of putting it in perspective with each socio-cultural context and heterogeneity of national and regional public policies carried out to control the COVID-19 pandemic. First, the questionnaire is supposed to be based on two “familiar” psycho-social models (COM-B and HBM) for explaining health behaviors (including adherence to official public health recommendations), while there is a huge body of literature in the field of HIV prevention (as well as other diseases) that has emphasized the limitations of such models for better understanding health behaviors and there does not seem any efforts to take that into account in the current ICARE protocol. Second, most of collected data come from convenience internet samples with significant decrease in the number of respondents as well as countries covered between the first and the fourth waves that have been completed. There is no guarantee in the way the described protocol will be able to determine the extent to which inter-countries variance (as well as intra-country variance through time) that would be observed effectively identify “real world” differences rather than heterogeneity in data collection, especially with such high number of targeted countries. The authors mention that internet panel data on representative samples of populations (stratified according to age, sex, province/region, education level, and income) have been obtained in four countries (Canada, Australia, UK and Ireland) with opportunities of further extension to a few other countries. It would have been a lot more appropriate to focus on such sampling method n the price to pay would have been to limit the number of targeted countries; although such surveys have intrinsic limitations of internet data collection, control of biases is easier and comparisons would have been more robust. Of course, papers based on analysis of data would allow reviewers to effectively assess the added value and validity of the ICARE survey but at this stage publishing a paper on its protocol does not yet seem appropriate.
--	--

REVIEWER	Dan Wu LSHTM
REVIEW RETURNED	19-Dec-2020

GENERAL COMMENTS	I think the study will have potentialI have included my detailed comments in the PDF file attached. - The reviewer provided a marked copy with additional comments. Please contact the publisher for full details.
--

VERSION 1 – AUTHOR RESPONSE

Reviewer 1

The authors of this paper present a study protocols for an international assessment of the link between COVID-19-related attitudes, concerns and behaviours in relation to public health policies. I have a few suggestions that you might want to consider:

- The paper should provide technical validation methods which will be used in study

We appreciate the reviewer's comment here; however, we are not 100% sure of the specific aspect of validation that the reviewer is referring to. For the representative samples, we have noted that weightings will be used (see page 11). We have also included brief details about our plan to determine the psychometric properties of the various components of the survey (see page 13). Finally, we added details about how the representative samples can be leveraged to identify any potential biases in the global sample, along with potential weighting strategies for the global samples if there are sufficient sample sizes in any one country (see page 14).

- The Figure 1b seems lack of the block connect to "Likelihood of adhering"?

It would seem that when the figure files uploaded, that a black background was incorporated which meant that the full figures were not completely visible. We have corrected this so that the figure should be completely viewable now.

- The survey documents at <https://doi.org/10.17605/OSF.IO/H8RW2> was deleted.

We apologise for this oversight. The iCARE documentation was updated after submission of the initial manuscript to include more comprehensive information. All updated documentation can now be found at: <https://osf.io/nswcm>, which is a permanent link.

Reviewer 2

In the case of HIV infection, thousands of Knowledge-Attitudes-Beliefs & Practices (KABP) surveys have been carried out worldwide in representative samples of the general population of multiple countries at various points in time and have been often very useful to guide public health authorities tailoring appropriate prevention messages and policies. The ICARE study is a Canadian-initiated international cross-sectional KABP survey in the context of the on-going pandemic of COVID-19 with the aim of allowing comparison at multiple points in time (through sequential waves) between samples in circa 140 countries. The present paper deals with the study protocol as well as some very preliminary data about the ongoing data collection process.

We appreciate the positive view of the reviewer about the potential of the iCARE study to provide benefit to the continued efforts to reduce the impact of the COVID-19 pandemic.

However, there are a number of limitations in the study protocol that raises doubts about the capacity of future analyses to fulfill this complex goal of valid multi-countries' comparison with the additional ambition of putting it in perspective with each socio-cultural context and heterogeneity of national and regional public policies carried out to control the COVID-19 pandemic.

First, the questionnaire is supposed to be based on two "familiar" psycho-social models (COM-B and HBM) for explaining health behaviors (including adherence to official public health recommendations), while there is a huge body of literature in the field of HIV prevention (as well as other diseases) that has emphasized the limitations of such models for better understanding health behaviors and there does not seem any efforts to take that into account in the current ICARE protocol.

Firstly, it is important to note that a unique feature of the iCARE study is that it was designed to be a **theoretically-driven survey**, which is an important departure from the abundance of a-theoretical survey studies on COVID-19 which have and are been conducted.

As indicated by the reviewer, no single model will be able to provide a complete understanding of the behaviours under study (which is arguably true of every model in science). In the context of health behaviors, there are at least 83 models/theories of behaviour change (see, *Michie et al, ABC of Behaviour Change Theories: An Essential Resource for Researchers, Policy Makers and Practitioners, 2014*), all of which have advantages and disadvantages. Further, our rationale for selecting both the COM-B and HBM is that they assess complementary constructs that are specifically relevant to the behaviours under study: adherence to public health prevention measures in the context of infectious disease prevention. Specifically, the *COM-B Model*^{1,2} predicts that behaviour change (and by extension, adherence to COVID-19 mitigation behaviours) depends on: (1) awareness of the (COVID-19) prevention measures and the ability to engage in those behaviours (capability); (2) the belief that behavioural prevention measures are personally relevant and important (motivation), and (3) having the social and environmental resources needed to adopt the behaviour (opportunity).² The *COM-B Model*, which is part of the *Behaviour Change Wheel*,³ is currently one of the most widely adopted behaviour change intervention models in the world, as it provides a comprehensive framework for both understanding behaviour and how to change it on an individual, societal, or systems level.^{2,4} We also designed our survey to assess the constructs in the *Health Beliefs Model (HBM)*^{5,6} which posits that in the context of adopting disease prevention measures, a person's belief in the personal threat(s) posed by an illness or disease, together with a person's belief in the importance and effectiveness of recommended behaviours, will predict the likelihood a person will (or will not) adopt a particular behaviour. This model has been most successful at predicting behaviour.^{7,8} Specifically, the theory allows for the capture of variables that are predictive/important because the constructs overlap heavily with other theories that are commonly used.⁹ Therefore, especially when used with the COM-B, drawing inspiration from the HBM helps us be extensive in our coverage so that people working with a wide variety of more specific models can find our data useful.

Of note, the iCARE study includes more than 190 international collaborators, many of which are world-renowned experts in behaviour change and behavioural medicine. As such, we selected our models after consultation with our international experts, which include Susan Michie, Robert West, Michael Vallis, Kenneth Freedland, Molly Byrne, Justin Pesseau, Tavis Campbell, and many others. In addition, the COM-B is ideal for the purpose of our study as it unifies ideas from a large number of disciplines within a unitary framework that everyone can use to communicate.

With regards to the question of the use of the COM-B and HBM in the context of HIV, given that this is not an area that we directly work in, we did a brief exploration of the literature to understand the prevailing controversies of these models in that field. Overall, we found that the HBM has been used extensively in the HIV literature and has been seen as a useful tool to explain a number health behaviours relevant to HIV. In contrast, we found very little HIV research that has explored the use of the COM-B (or behaviour change wheel, which is an extension of the COM-B). Overall, these studies seemed to indicate positive results for the COM-B to explain behaviours or mechanisms of action for interventions targeting behaviour change. Interestingly, the critics of the use of current health behaviour models (including the HBM) in the context of HIV seem to centre around two issues. The first issue is that they do not account for the powerfully motivating factors of sexual desire and pleasure, which has little relevance in the context of COVID-19. The second issue is that they argue that these models focus only on individual-level factors with little consideration of any other level of influence on behaviour (e.g., environmental or social factors). Whilst this latter issue is true of the HBM, it is not for the COM-B, which considers an array of factors (including environmental and social) which influence behaviour, and is precisely why we used both models within the context of the iCARE study. Below are some of the articles that reflect the arguments made above:

- Rosenstock I.M., Strecher V.J., Becker M.H. (1994) The Health Belief Model and HIV Risk Behavior Change. In: DiClemente R.J., Peterson J.L. (eds) Preventing AIDS. AIDS Prevention and Mental Health. Springer, Boston, MA. https://doi.org/10.1007/978-1-4899-1193-3_2
- King, Sexual behaviour change for HIV: where have theories taken us? Geneva: UNAIDS Best Practise Collection, 1999.
- Simoni JM, Ronen K, Aunon FM. Health Behavior Theory to Enhance eHealth Intervention Research in HIV: Rationale and Review. *Curr HIV/AIDS Rep.* 2018 Dec;15(6):423-430. doi: 10.1007/s11904-018-0418-8. PMID: 30511186; PMCID: PMC6324197.
- Michielsen K, Chersich M, Temmerman M, Dooms T, Van Rossem R. Nothing as Practical as a Good Theory? The Theoretical Basis of HIV Prevention Interventions for Young People in Sub-Saharan Africa: A Systematic Review. *AIDS Res Treat.* 2012;2012:345327. doi: 10.1155/2012/345327. Epub 2012 Aug 1. PMID: 22900155; PMCID: PMC3415137.
- Witzel TC, Weatherburn P, Bourne A, Rodger AJ, Bonell C, Gafos M, Trevelion R, Speakman A, Lampe F, Ward D, Dunn DT, Gabriel MM, McCabe L, Harbottle J, Moraes YC, Michie S, Phillips AN, McCormack S, Burns FM. Exploring Mechanisms of Action: Using a Testing Typology to Understand Intervention Performance in an HIV Self-Testing RCT in England and Wales. *Int J Environ Res Public Health.* 2020 Jan 10;17(2):466. doi: 10.3390/ijerph17020466. PMID: 31936798; PMCID: PMC7014239.

Second, most of collected data come from convenience internet samples with significant decrease in the number of respondents as well as countries covered between the first and the fourth waves that have been completed. There is no guarantee in the way the described protocol will be able to determine the extent to which inter-countries variance (as well as intra-country variance through time) that would be observed effectively identify “real world” differences rather than heterogeneity in data collection, especially with such high number of targeted countries. The authors mention that internet panel data on representative samples of populations (stratified according to age, sex, province/region, education level, and income) have been obtained in four countries (Canada, Australia, UK and Ireland) with opportunities of further extension to a few other countries. It would have been a lot more appropriate to focus on such sampling method n the price to pay would have been to limit the number of targeted countries; although such surveys have intrinsic limitations of internet data collection, control of biases is easier and comparisons would have been more robust.

The reviewer's point is well taken and we agree that with infinite funds, extensive representative sampling would have been the ideal way to obtain a comprehensive view of the behavioural perspectives related to COVID-19. Unfortunately, in spite of the importance of behaviour in the context of COVID-19 (including people getting the new vaccines), it has been estimated that less than .005% of research (worldwide) for COVID-19 has been in the behavioural or social sciences (www.bessi.net.au). As indicated in the manuscript, we have been fortunate enough to be able to obtain some funding, though this has not been enough to do the mass representative sampling that the reviewer is suggesting. As such, we have had to rely on other methods and mechanisms for widespread data collection. Most of the recruitment efforts for the global sample are being done by the local collaborators within a country (supported by the central study group who are providing materials to distribute) and we continue to be impressed by the collective efforts of our international collaborators, who have come together to undertake such a project and committed so many hours 'for free.' The consequence of such an egalitarian approach is that it is not possible to limit the participation of certain countries. However, it should be noted that using this approach, global sampling has yielded high numbers of responses from several key countries, for example Canada (n= 15,406), Italy (n= 2,042), US (n= 2,232), Colombia (n= 2,301), as well as, France (n= 3,883), Taiwan (n= 2,597), Brazil (n= 2,297), Kenya (n= 865), Israel (n= 1,640), and Turkey (n= 1,423), which will allow for in-depth analyses of COVID-19 policy impacts across a range of countries. It should also be noted that this convenience data is still useful for drawing conclusions about general relationships between certain variables (e.g., correlations between beliefs and behaviors) that are not tied to specific countries, and may possibly even offer a more diverse sampling for this purpose than representative samples from any given country. Not all inferences require representative samples, and some are better served by purposive samples (for which achieving diversity is key to approximating a general response).

To our knowledge, this is one of the largest, longest (in terms of time), and most comprehensive COVID-19 related behavioural studies to be undertaken. That being said, we agree that the variability in responses overtime and across countries means that we will not be able to provide comprehensive coverage and results from all the countries from which data is captured. In spite of this limitation, it is notable that we do have excellent coverage in multiple countries (as described above), across different continents, and over time. As indicated in the manuscript, analyses will ideally be at "the level of country, but for those with limited data it might be at the level of continent or for those with large amounts of data it might be at the level of region."

Of course, papers based on analysis of data would allow reviewers to effectively assess the added value and validity of the ICARE survey but at this stage publishing a paper on its protocol does not yet seem appropriate.

We thank the reviewer for their comment, but would like to refer them to the opening description of the BMJ Open's requirements for a protocol paper (and clarify that our study is currently ongoing and planned to continue at least through the end of 2021):

"Protocol manuscripts should report planned or ongoing research studies. If data collection is complete, we will not consider the manuscript. We encourage the submission of protocol manuscripts at an early stage of the study. Protocols nearing completion of data collection will be treated on a

case by case basis and the final decision on whether to consider a protocol for publication will rest with the Editor.

Publishing study protocols enables researchers and funding bodies to stay up to date in their fields by providing exposure to research activity that may not otherwise be widely publicised. This can help prevent unnecessary duplication of work and will hopefully enable collaboration.”

We believe the current manuscript meets this requirement and that the need to publish papers would seem to be implicitly discouraged in this description. As indicated above, we would like to also highlight the fact that we will be doing analyses to evaluate the strength of our methods, but, consistent with similar large scale studies, these evaluations will require entire papers of their own.

Reviewer 3

I think the study will have potential. I have included my detailed comments in the PDF file attached.

We thank the reviewer for their positive comments. The responses to the detailed comments are provided below:

1. We have changed the comment that there is current no vaccine to indicate that there have been recently developed vaccines (pages 4 and 7).
2. We have highlighted the specific sub-groups that we will initially focus on (page 8).
3. With regards to the economic growth, the structure of the iCARE study means that we have the capacity to incorporate a variety of additional data sources beyond just the survey (for example the Oxford Policy data and the John Hopkins cases/mortality data). Though we don't have country-level economic data included in the plan, if an appropriate data source can be found this could be added at a later date.
4. The health, economic, and quality of life components are primarily included in the impacts elements of the survey. This has now been made explicit (see page 10).
5. As indicated in the responses to reviewer 2, the nature of the recruitment means that there are no specific target sample sizes for each country.
6. Fuller details of both the representative and convenience samples are provide on pages 10 and 11.
7. Given the global urgency to understand the COVID-19 pandemic, we were as inclusive as possible for the collaborators. The current list of collaborators cover internationally recognised experts on variety of topics including behaviour change, public health, infectious disease, biostatistics, etc., through to graduate students in these fields. With regards to the reduction of bias in the convenience sample, as indicated in the response to reviews for reviewer 1, where possible we will leverage the representative samples to explore bias and weightings and where there is sufficient sample we will be able to weight the data to provide an approximation for the country (see page 14).
8. We have expanded our description of the way in which the recruitment of the global sample is conducted (see pages 10 and 11).
9. The reviewer raises an excellent point about accessing more marginalised groups. It should be noted that a key limitation to the survey is that it does need to be conducted online which unto itself will exclude certain marginalised groups. Centrally, there was no specific targeting of any particular groups, though local collaborators are being encouraged to reach as many diverse

individuals as possible. That being said, of note, one unique and important aspect of the survey is that it is available in 36 languages, which means that certain marginalised groups (e.g., immigrants to Western countries, like Canada, the US and France and which are highly represented) will likely be able to complete the survey in their native language. We see this as a key way in which our study is inclusive.

10. The representative sampling is primarily a function of funding. At a secondary level, there is an attempt to target countries in different continents, at different stages/phases of the pandemic, with varying perspectives on managing the pandemic within their countries, or those with more participants in the convenience sample. For example, in a funding application currently under review, the following countries are being targeted for representative sampling: Canada (the host country for the iCARE study and a North American country which is currently in a notable second wave); Australia (an Oceania country which has managed to successfully manage both a first and second wave, but is currently in a third wave); Ireland (a European country which has had moderate success in managing both waves); the United States (a North American country which is leading the world in COVID-19 cases and deaths); Italy (a European country which was the original epicentre of the first wave in Europe and has the most COVID-19 deaths in Europe but is currently on a downward trend from the second wave); and Colombia (a LMIC South American country which has had significantly fewer deaths compared to cases than most other countries in the continent).
11. We have provide a specific example of one of the representative samples to the current manuscript (see page 11).
12. With regards to the additional data sources, our original manuscript was focused more on the specifics of the survey as that is unique to the iCARE study. However, we have provided one example of how this will be combined with the other data sources to contextualise the use of a variety of data elements (see page 14).
13. The reviewer has noted that we were able to reach more countries than just the 40 covered by the collaborators. This is primarily due to the fact that some languages cover more that one country (e.g., English, French, and Spanish) and that a number of the collaborators are connected to some impressive international networks. This is a strength of the current iCARE study and the impressive array of collaborators who are participating in the study. The issue of the recruitment strategy is covered in the responses to reviewer 2.
14. The issue of the nature of recruitment and the numbers over time are covered in the responses to reviewer 2.
15. Further details on the sampling methods for the representative samples are provided on page 11. With regards to the question of overlapping individuals, each survey that is completed is anonymous and for most representative samples, the participants are drawn from a large pool of individuals (e.g., for Canada Leger has a panel of over 400,000 Canadians from which the sample of 3,000 is drawn for each survey round). As such, it is theoretically possible that some individuals could complete more than one survey, though, there is a very low probability that this would occur.
16. We appreciate the reviewers' comments around the vagueness of the analysis section. As indicated in this section, the complexity of the data and the breadth of possible questions that can be answered by the data within the iCARE study means that we can't be specific about all potential analyses that will eventually be undertaken (partly due to the forever evolving nature of the pandemic). That being said, we have tried to provide an example based on one of the key outcomes (see page 14), which are detailed earlier in the manuscript (see page 8). The issue of how analyses have been envisaged in those countries where the sample sizes are on the lower side has been addressed below in point 18.
17. We are pleased to say that since the original submission of this manuscript, we have been able to make considerable progress in providing results and information to a variety of stakeholders. We have now updated the Interpretation section (page 15) to include the websites where current results can be found and some of the governments that we have provided data to (e.g., Canada, Ireland, Australia). We haven't provided details of specific results as these will have or will be included in specific publications on those topics. Also, given that this is part of a larger package of information that governments have used to make decisions, it is difficult to say that the iCARE data was the sole generator of social and policy impact.
18. In an ideal world, it would be fantastic to have extensive responses from each country at each survey round. Unfortunately, the nature of the study precluded us from being able to guarantee such numbers. However, as indicated in the manuscript, where there are insufficient numbers in any country then appropriate collapsing of data across countries to get continent level data will be

considered. Alternatively, in situations where a country has significant amounts of data it would be possible to provide more granular data and provide information based on regions.

Additional References

1. Michie S, van Stralen M, West R. The Behavior Change Wheel: a new method for characterizing and designing behavior change interventions. *Implementation Science*. 2011;6:42.
2. West R, Michie S, Rubin G, Amlot R. Applying principles of behaviour change to reduce SARS-CoV-2 transmission. *Nature Human Behaviour*. 2020;<https://doi.org/10.1038/s41562-020-0887-9>.
3. Michie S, Atkins L, West R. *The Behavior Change Wheel: A guide for designing interventions*. London: Silverback Publishing.; 2014.
4. West R, Michie S, Atkins L, Chadwick P, Lorencatto F. Achieving Behaviour Change: A Guide for Local Government and Partners. *Public Health England*. 2020.
5. Rosenstock I, Strecher V, Becker M. Social learning theory and the health belief model. *Health Education Quarterly*. 1971;15(2):175-183. .
6. Rosenstock I. The health belief model and preventive health behavior. *Health Education Monographs*. 1974;2:354-386.
7. Jones C, Smith H, Llewellyn C. Evaluating the effectiveness of health belief model interventions in improving adherence: a systematic review. *Health Psychology Review*. 2014;8:253-269.
8. Sulat J, Prabandari Y, Rossi S, al. e. The Validity of Health Belief Model Variables in Predicting Behavioral Change. *Health Education*. 2018;118:499-512.
9. Sheeran P, Klein WM, Rothman AJ. Health Behavior Change: Moving from Observation to Intervention. *Annu Rev Psychol*. 2017;68:573-600.

VERSION 2 – REVIEW

REVIEWER	Viet-Phuong La Phenikaa University, Hanoi, Vietnam
REVIEW RETURNED	25-Jan-2021

GENERAL COMMENTS	I am happy with the changes that the authors have made.
---

REVIEWER	Pr MOATTI Jean Paul Aix Marseille University (South Eastern France)
REVIEW RETURNED	01-Feb-2021

GENERAL COMMENTS	This new version of the Bacon et al. paper on the ICARE study presents some improvements compared to the previous draft that clearly attempt at taking into account reviewers' concerns. In particular, authors recognize that the main source of data – on line surveys using the capture method through snowbal sampling – « is likely to create some bias in the sample ». Unfortunately, the authors remain insufficiently precise about the way they will deal with the major limitations of such sampling method when the analysis has the ambition to provide robusqt multi-countries comparison. As mentioned in a previous review, there is no guarantee in the way the described protocol will be able to determine the extent to which inter-countries variance (as well as
--

	intra-country variance through time) that would be observed effectively identify “real world” differences rather than heterogeneity in data collection, especially with such high number of targeted countries. The authors mention the opportunity that “in countries where there is sufficient data in the convenience sample, (they)will apply weights to allow the data to provide national approximations ». Why this approach is not systematic using census data as the reference ? Is this because the composition of some samples would imply so important weights that extrapolation would not have any meaning ? The authors would be more convincing if they had provided some information about the age, gender and socio-economic characteristics of their respondents and the exten to which they depart in each country from the global structure of the adult population. In addition, pretending that some bias « can be adjusted using weightings from the representative samples that are being collected » (for the moment in four countries –Canada, Australia, UK and Ireland) seems quite unrealistic (and even for these countries pooling data from these two different sampling methods does not seem appropriate). In a similar vein, it remains unclear what would be the goal in the statistical analysis of using « multiple imputation mezhods for missing data ». Would it be only applied for completing missing responses of individuals’ questionnaires or otherwise ? In addition, authors remain unclear about the wa y they will take into account the decreasing number of respondents in the 6 consecutive waves of their international convenience sample (from nearly 29,000 respondents in 137 countries at round 1 of April 2020 to only circa 2,500 individuals from 73 countries at round 6 in November 2020). Suggesting, more or less explicitly, that « generalised linear models will be developed to estimate systematic differences » and account for biases in these heterogeneous convenience samples is clearly inappropriate. The authors keep on claiming that one strenght of their compariove study is that the questionnaires are rooted in two well-known psycho-socialk behavioural models (COM & HBM) and it is true that referring to explicit theoretical frameworks for building a questionnaire is methodogically sound. However, they still do not recognize the well-established (in the literature on health and prevention) limitations of such models to understand actual attitudes and behaviors especially in the context of a major crisis characterized by systematic uncertainties in both knowledge and policies. Therefore, it would be better that authors submit papers with some prelmùinary analysis of their empirical date may be with a more limited scope focusing on robust data for compataive purpose between countries.
--	---

REVIEWER	Dan Wu London School of Hygiene and Tropical Medicine
REVIEW RETURNED	26-Jan-2021

GENERAL COMMENTS	Thanks for revising the paper. The authors have properly addressed my concerns. I am happy to recommend publication.
--

VERSION 2 – AUTHOR RESPONSE

Response to reviews

Reviewers 1 and 3

We are pleased to see that both reviewers 1 and 3 are very satisfied with the extensive edits that we made to the initial submission and requested no further edits to the current version of the manuscript.

Reviewer 2

This new version of the Bacon et al. paper on the ICARE study presents some improvements compared to the previous draft that clearly attempt at taking into account reviewers' concerns. In particular, authors recognize that the main source of data – on line surveys using the capture method through snowball sampling – « is likely to create some bias in the sample ».

We appreciate the reviews acknowledgement of the work and effort we put into addressing all the reviewers' comments.

Unfortunately, the authors remain insufficiently precise about the way they will deal with the major limitations of such sampling method when the analysis has the ambition to provide robust multi-countries comparison. As mentioned in a previous review, there is no guarantee in the way the described protocol will be able to determine the extent to which inter-countries variance (as well as intracountry variance through time) that would be observed effectively identify “real world” differences rather than heterogeneity in data collection, especially with such high number of targeted countries.

The authors mention the opportunity that “in countries where there is sufficient data in the convenience sample, (they)will apply weights to allow the data to provide national approximations». Why this approach is not systematic using census data as the reference ? Is this because the composition of some samples would imply so important weights that extrapolation would not have any meaning ? The authors would be more convincing if they had provided some information about the age, gender and socio-economic characteristics of their respondents and the extent to which they depart in each country from the global structure of the adult population. In addition, pretending that some bias « can be adjusted using weightings from the representative samples that are being collected » (for the moment in four countries –Canada, Australia, UK and Ireland) seems quite unrealistic (and even for these countries pooling data from these two different sampling methods does not seem appropriate). In a similar vein, it remains unclear what would be the goal in the statistical analysis of using « multiple imputation methods for missing data ». Would it be only applied for completing missing responses of individuals' questionnaires or otherwise ? In addition, authors remain unclear about the way they will take into account the decreasing number of respondents in the 6 consecutive waves of their international convenience sample (from nearly 29,000 respondents in 137 countries at round 1 of April 2020 to only circa 2,500 individuals from 73 countries at round 6 in November 2020). Suggesting, more or less explicitly, that « generalised linear models will be developed to estimate systematic differences » and account for biases in these heterogeneous convenience samples is clearly inappropriate.

It would seem that the reviewer's main preoccupation is still around aspects of the methodological design of the study, particularly regarding our use of snowball sampling. As we have acknowledged in our response to the previous reviews, we agree that if we had infinite funds we would have conducted multiple longitudinal representative samples which would have allowed us to undertake different kinds of analyses and taken a different perspective on how to manage the data. However, this would have required millions of dollars (each representative sample costs on average \$20,000CAD or more), funding for which is not available. That being said, we would like to reiterate that this study did receive peer-reviewed funding from several countries (e.g., Canada, Ireland, Australia) and this protocol paper reflects what was proposed and funded by three independent agencies, attesting to the scientific quality and feasibility of the project. Of note, these methods have been used by others and have resulted in several high quality publications (see Kowal et al, *Health and Well Being*, 2020; Yamada et al, *Scientific Data*, 2021; and Lieberoth et al, *Royal Society Open Science*, 2021), further attesting to the scientific acceptability and potential impact of this work.

Due to the constantly evolving nature of the pandemic, there are number of aspects related to the project that cannot be explicitly planned a-priori, including the sample sizes that will be obtained from snowball sampling from one time period to the next. We also cannot predict the results of funding applications that have, and will be, submitted to enhance the number of representative samples that we will be able to capture. These aspects have and will continue to have impacts on various aspects of the statistical analyses that can and will be conducted. What we have provided is a general overview of the statistical perspective that we will take. However, any specific question or request from a government for information will need its own specific analytical plan. For example, if someone, e.g., a Governmental Department of Public Health, requests a brief snapshot of a specific situation (e.g., attitudes towards testing and contact tracing among young people), then a simple descriptive analysis or correlational analysis would be conducted. In contrast, if a Government Department or Agency were interested in a more complex analysis of an evolving series of contingent factors on a specific outcome (e.g., how various policy impacts are affecting adherence to different prevention measures over time as a function of age and sex), then a multi-level modelling approach would likely be appropriate. To reflect this, we have added a line to the statistical analysis section identifying that "With a study of this magnitude, it is impossible to detail all possible analyses that could be conducted, as these will vary based on the specific questions that might be received from governments or researcher partners. However, the following section provides a high-level overview of the kinds of 'basic' analytical strategies that will be conducted with the data."

With regards to the issue of weighting, it would seem that the reviewer may be confounding two different issues. In the data harmonisation section, we have indicated "In countries where there is sufficient data in the convenience sample, we will apply weights to allow the data to provide national approximations (23-25)." The references that we cited are articles that provide standard guidance about the use of national statistics to weight data. We reported this information this way for brevity, as these can be quite complex from country to country, and this is not the primary focus on the iCARE study. Part of the issues with this kind of approach is to ensure that there is sufficient data to be able to do such weighting. Given the nature of snowball sampling, we can't guarantee the number of participants we may get from any particular country. For example, there are a few countries where we have less than 50 respondents and I am sure that the reviewer is not advocating for us to try and apply national weightings to this small group, hence why we included the caveat that we would only apply this to those with sufficient data.

In a separate weighting issue, the fact that we are collecting representative samples provides us with the potential to look at how the representative samples are responding to questions relative to the comparable country convenience samples, which may provide insight into potential bias that snowball sampling creates. Using this information, it is possible that further weighting of the convenience sample could occur. Of note, since the submission of the last revision, several members of our team are looking at this highly original and complex issue with the intention of conducting empirical research on appropriate ways to leverage representative samples to inform non-random data capture samples. As a result, we have modified the sentence describing this in the manuscript to more clearly reflect this: "These analyses may provide insights into potential areas of bias and so that potential further weightings could be applied to the global sample."

For the issue of missing data, the same issues of the complexity of the data mean that there isn't a standard response that we could blankly apply to all the data. This is why we highlighted the need to be cautious in this approach, stating "Patterns of missing data will be examined and, where appropriate, accounted for by using multiple imputation techniques (21, 22)." To broadly apply missing data imputations is contrary to best practices around missing data, and the exploration of the nature of the missingness is key in making decisions around how to move forward.

The authors keep on claiming that one strength of their comparative study is that the questionnaires are rooted in two well-known psycho-social behavioural models (COM & HBM) and it is true that referring to explicit theoretical frameworks for building a questionnaire is methodologically sound. However, they still do not recognize the well-established (in the literature on health and prevention) limitations of such models to understand actual attitudes and behaviors especially in the context of a major crisis characterized by systematic uncertainties in both knowledge and policies.

We have to be transparent and say it is unclear what the reviewer is trying to convey here. In our previous response, we spent a lot of time detailing why developing a **theoretically-driven survey** was critically important and highly unique (I am sure that the reviewer would not be suggesting that an atheoretical or completely random series of questions would be preferable). In fact, the reviewer seems to acknowledge that building questionnaires based on explicit theoretical frameworks is indeed "methodologically sound". Furthermore, we provided numerous examples where both the COM-B and the HBM have shown benefit (both within the broad health literature and within the area of HIV). We also highlighted that, as with every facet of science (and this might be particularly true of the behavioural sciences, which seek to predict highly complex behaviours that are influenced by multiple factors), that no single theory is likely to account for every aspect of a subject under study. However, we'd like to reassure the reviewer that we selected these 2 theories for the current project after extensive consultation with several internationally renowned health behaviour change experts including the authors of the theories themselves (e.g., Dr. Susan Michie at UCL is arguably the top behavioural health expert in the world as is the author of the COM-B model and Behaviour Change Wheel, and is a collaborator on the iCARE project). We hope the reviewer finds this response satisfactory.

Therefore, it would be better that authors submit papers with some preliminary analysis of their empirical data may be with a more limited scope focusing on robust data for comparative purpose between countries.

While we appreciate that the reviewer may want to see some data, we would like to remind the reviewer of BMJ Open's requirements for a protocol paper (and clarify that our study is currently ongoing and planned to continue at least through the end of 2021):

"Protocol manuscripts should report planned or ongoing research studies. If data collection is complete, we will not consider the manuscript. We encourage the submission of protocol manuscripts at an early stage of the study. Protocols nearing completion of data collection will be treated on a case by case basis and the final decision on whether to consider a protocol for publication will rest with the Editor.

Publishing study protocols enables researchers and funding bodies to stay up to date in their fields by providing exposure to research activity that may not otherwise be widely publicised. This can help prevent unnecessary duplication of work and will hopefully enable collaboration."

We believe the current manuscript meets this requirement and that the need to publish papers would seem to be implicitly discouraged in this description. As indicated above, we would like to also highlight the fact that we will be doing analyses to evaluate the strength of our methods, but, consistent with similar large-scale studies, these evaluations will require entire papers of their own.